# How Social Support Affects Resilience in Disadvantaged Students: The Chain-Mediating Roles of School Belonging and Emotional Experience

**DOI:** 10.3390/bs14020114

**Published:** 2024-02-04

**Authors:** Zhenyu Li, Qiong Li

**Affiliations:** 1Center for Teacher Education Research, Faculty of Education, Beijing Normal University, Beijing 100875, China; lizhyu@mail.bnu.edu.cn; 2Academy of Plateau Science and Sustainability, People’s Government of Qinghai Province & Beijing Normal University, Xining 810016, China

**Keywords:** social support, school belonging, emotional experience, resilience

## Abstract

This study aims to utilize data from the Programme for International Student Assessment (PISA), 2018, conducted in four provinces and cities in China, to investigate the impact of social support on the resilience of disadvantaged students. It specifically focuses on the chain-mediated effects of school belonging and emotional experiences. To achieve this, the study selected 2997 disadvantaged students as participants and employed path analysis to examine the mediating effects. The results indicate that teacher support, parental support, school belonging, and positive emotional experiences significantly positively influence the resilience of disadvantaged students, while fear of failure significantly negatively affects resilience. Additionally, teacher and parental support indirectly impact the resilience of disadvantaged students through the chain-mediated effects of school belonging and positive emotional experiences. Similarly, teacher and parental support also influence the resilience of disadvantaged students through the chain-mediated effects of school belonging and fear of failure. To enhance the resilience development of disadvantaged students, it is recommended that teachers and parents provide active attention and support to these students. Simultaneously, there should be increased focus on the psychological well-being of disadvantaged students by strengthening their mental health education.

## 1. Introduction

Adverse circumstances encompass a spectrum of risk factors within the family environment that detrimentally affect children’s psychological and social adaptation, such as parental divorce, parental conflicts, and negative parent–child interactions [1]. Students living amidst adverse circumstances are labeled as disadvantaged. The convergence of adverse influences stemming from family dynamics, school education, socio-economic factors, and cultural elements not only diminishes the academic accomplishments of disadvantaged students in comparison to their more privileged counterparts, but also gives rise to challenges in their self-concept, socio-emotional disposition, peer relationships, and societal behavior [2]. Moreover, it affects the future prospects of academic advancement, employment, and social integration for disadvantaged students, undermining their potential to break societal class barriers and impeding upward social mobility, significantly obstructing societal progression [3]. Some studies indicate that the impact of adverse circumstances on children’s development accumulates over time, leading to an escalation of inequality gaps as children age, especially during secondary education [4,5]. Therefore, fostering the healthy growth and comprehensive development of disadvantaged students is pivotal in achieving educational equity and serves as a critical factor for marginalized groups to attain social mobility while disrupting the intergenerational transmission of poverty.

In the development of disadvantaged students, the cultivation of their resilience is crucial. Resilience denotes the ability to adapt successfully to internal and external pressures [6,7], enabling individuals to confront adversities and achieve positive adjustments and growth in their environment [8,9,10]. Students with higher levels of resilience often demonstrate greater proactive adaptation and control over adverse factors, exhibiting remarkable traits in self-adjustment, self-identity, and the establishment and maintenance of interpersonal relationships [11]. As research advances, the study of resilience has transitioned from a dualistic approach to a more systemic one based on the interaction between individual and external environments, shifting the focus from outcomes to processes. Researchers have begun viewing resilience as a positive adaptation process within adversity, considering it a continuous interaction between internal and external environments [12]. Among external factors, social support is believed to play a crucial role in fostering resilience development. Social support is widely regarded as a key factor in coping with and overcoming difficulties [13,14,15,16], showing significant correlation with an individual’s emotional regulation, life satisfaction, and resilience [17,18,19,20,21]. Studies also suggest an intrinsic connection between social support and resilience, generally positing that social support fosters the development of individual resilience [22]. Within individual factors, students’ sense of school belonging and emotional experiences also play pivotal roles in their resilience development [23,24,25]. For instance, research has shown that school belonging contributes to preventing anxiety and depression [26,27], enhancing resilience [23], and boosting self-esteem [28], positively impacting students’ psychological well-being [29].

However, in investigating the factors influencing individual resilience development, most studies have primarily focused on environmental or individual factors independently, with little integration of these two major factors for comprehensive consideration [13,25]. Additionally, prior research has inadequately explored the mechanisms affecting individual resilience development, often failing to consider potential relationships among influencing factors [23]. Therefore, this study aims to combine environmental factors influencing resilience development with individual factors, intending to explore the impact mechanisms of social support on the resilience development of disadvantaged students. Simultaneously, it seeks to elucidate the roles of school belonging and emotional experiences in this context, aiming to provide insights into promoting resilience development among disadvantaged students.

## 2. Literature Review

### 2.1. Social Support and Resilience: Direct Association

Social support refers to the assistance an individual receives from external sources (such as family, friends, or community organizations) in various forms, including material, emotional, and informational aid, within stressful circumstances [30]. Previous research has identified social support as a protective factor for children’s mental health [14]. Level of social support can predict the mental and physical well-being of children. Strong social support (from both parents and peers) correlates with improved psychological development and social adaptation in children, leading to reduced levels of loneliness and depression [31,32] and higher self-esteem and sense of self-worth [33,34]. More significantly, social support constitutes a crucial external influence in shaping an individual’s resilience [35]. A plethora of studies confirm social support as a key protective factor for students’ resilience [36,37]. For disadvantaged students, support from society, family, and school is critically needed. Positive family relationships and an encouraging family atmosphere can encourage adolescents to exhibit resilient behaviors [38]. Additionally, positive relationships with teachers and peers can enhance children’s resilience [39]. Furthermore, research indicates a stable and systemic relationship exists between students’ social support and resilience. As students build their social networks (including support from family, friends, and school), their abilities to cope with setbacks and recover from adverse situations tend to develop [40]. Therefore, based on the aforementioned literature review, the following hypothesis is proposed:

**Hypothesis** **1.**
*Social support significantly influences the development of resilience among disadvantaged students.*


### 2.2. The Mediating Role of School Belonging

School belonging refers to the extent to which students perceive themselves to be accepted, respected, and supported within the school environment [41], reflecting an emotional connection to their school experiences [42]. Belonging is one of the fundamental human needs that influences an individual’s cognition, emotions, and behaviors [43]. Adolescence marks a critical period for self-identity formation, socialization, and transitioning from childhood to adulthood [44,45]. School belonging correlates positively with positive self-identity [46], aiding in the construction of positive psychological and social adaptation [47]. Several studies have shown a positive correlation between school belonging and resilience [48,49]. Additionally, research indicates a significant positive direct impact of school belonging on resilience [50]. Schools provide adolescents with an inclusive environment that meets their growing need for belonging, thus facilitating the cultivation of positive coping abilities and resilience in adolescents [51].

On the other hand, Self-Determination Theory posits that when students feel satisfied with the support they receive from others, their autonomy and sense of belonging are fulfilled [52]. Studies suggest that parental, teacher, and peer support behaviors can predict students’ attitudes and sense of belonging at school [53]. Specifically, within supportive and inclusive learning environments, students experience high-quality interpersonal interactions and relationships, resulting in a positive sense of school belonging. Conversely, in unsupportive or threatening environments, students often feel marginalized and excluded, leading to a lack of school belonging [54]. Researchers conducted a meta-analysis summarizing individual- and social-level factors influencing school belonging [55]. The results emphasized social support as a crucial social factor influencing students’ sense of belonging at school, and parental and teacher support exerted a more substantial impact on students’ school belonging compared to peer support. Therefore, based on the aforementioned literature review, the following research hypotheses are proposed:

**Hypothesis** **2.**
*Social support significantly influences school belonging among disadvantaged students.*


**Hypothesis** **3.**
*School belonging significantly impacts the resilience of disadvantaged students.*


**Hypothesis** **4.**
*School belonging mediates the impact of social support on the development of resilience among disadvantaged students.*


### 2.3. The Mediating Role of Emotional Experience

Emotional experiences can be categorized into positive and negative emotional experiences. Individuals who frequently experience positive emotions tend to display higher levels of optimism, a calmer state of mind [56], and increased resilience [57]. Conversely, individuals experiencing frequent negative emotions are prone to psychological health issues such as anxiety, depression, and problematic behavior, leading to lower life satisfaction [58]. The experience of positive emotions, particularly in stressful situations, constitutes a crucial aspect of resilience acquisition. Fredrickson posits that positive emotions expand an individual’s momentary thought–action repertoire, broaden their scope of attention, enhance cognitive flexibility, build coping resources, and restore physiological stress responses to baseline levels [59]. It is through these functions of positive emotions that individuals develop resilience post-stress. Research has also found that negative emotions (such as anxiety and depression) have adverse effects on the development of resilience [60,61]. In other words, negative emotions can be considered as obstructive factors hindering the development of resilience in adolescents.

On the other hand, studies indicate that social support negatively predicts negative emotions like depression and anxiety among adolescents. Individuals with higher social support levels tend to exhibit lower levels of depression and anxiety [62]. The general benefits model of social support, also known as the main effects model, emphasizes that regardless of an individual’s stress state, social support meets their needs, provides a sense of security, fosters positive emotional experiences, and reduces psychological stress, thus promoting and benefiting an individual’s mental and physical health [63,64]. Moreover, research has highlighted that perceived social support has a protective effect on psychological adjustment, facilitating increased positive emotional experiences [65]. Therefore, based on the above literature review, the following research hypotheses are proposed:

**Hypothesis** **5.**
*Social support significantly influences the emotional experiences of disadvantaged students.*


**Hypothesis** **6.**
*Emotional experiences significantly impact the resilience of disadvantaged students.*


**Hypothesis** **7.**
*Emotional experiences act as a mediator in the impact of social support on the development of resilience among disadvantaged students.*


### 2.4. The Relationship between School Belonging and Emotional Experience

A longitudinal study conducted in the United States revealed that students with a stronger sense of school belonging tend to exhibit more optimism and emotional stability, while those with weaker feelings of belonging often experience heightened levels of depression and anxiety [66]. Other studies have found that whether school belonging is assessed as a whole or divided into several sub-dimensions, it remains associated with negative emotions like anxiety and depression among adolescents, serving as a crucial predictor of these emotions [67]. Particularly, adolescents with a stronger sense of school belonging may experience more positive emotions such as relaxation and contentment, while conversely, students with lower school belonging are more prone to negative emotions like loneliness and anxiety [68]. Therefore, based on the literature review presented above, the following research hypotheses are proposed:

**Hypothesis** **8.**
*School belonging significantly influences the emotional experiences of disadvantaged students.*


**Hypothesis** **9.**
*School belonging and emotional experiences serve as a sequential mediating mechanism in the impact of social support on the resilience of disadvantaged students.*


## 3. Materials and Methods

### 3.1. Data Source and Sample

The data used in this study were derived from the PISA 2018 survey conducted in four Chinese provinces and municipalities (Beijing, Shanghai, Jiangsu, and Zhejiang), comprising 12,058 approximately 15-year-old secondary school students from 361 schools. The sample information is presented in Table 1. The PISA 2018 survey employed a two-stage sampling method. Initially, over 150 schools were selected from regions with varying levels of educational development in each country. Subsequently, 35 approximately 15-year-old students were randomly selected from these schools to participate in the assessment. Given the focus of this study on disadvantaged students, it was necessary to filter the sample data. PISA defines students within the lower 25% range of the Index of Economic, Social, and Cultural Status (ESCS) in their county (region) as disadvantaged [69]. It is important to clarify that Economic, Social, and Cultural Status (ESCS) is derived from a composite score based on three indicators: highest parental occupation (HISEI), parental education (PAREDINT), and home possessions (HOMEPOS), which includes books in the home. This indicates that the adverse conditions experienced by socioeconomically disadvantaged students primarily stem from lower parental occupational status, lower parental educational attainment, and insufficient family wealth. Hence, based on this operational definition, a final selection of 2997 disadvantaged students was made for the study.

### 3.2. Variables and Measures

The variables utilized in this study were sourced from the PISA 2018 dataset. Detailed information regarding the items corresponding to the variables and their structure can be found in the PISA 2018 Technical Report (website: https://www.oecd.org/pisa/data/pisa2018technicalreport/ (accessed on 3 December 2023)).

#### 3.2.1. Independent Variable

*Social support.* In this study, social support is divided into two dimensions: teacher support and parental support. The teacher support dimension was measured by four items (e.g., teachers helping students with their studies) measured by frequency (1 (none or almost none) to 4 (every class)). For the validation of these four items, a reliability and validity test was conducted. The Cronbach’s alpha coefficient was found to be 0.849. The results of the confirmatory factor analysis indicated a good model fit: X^2^ = 46.339, *df* = 2; CFI = 0.991; TLI = 0.974; RMSEA = 0.086; SRMR = 0.015. Additionally, the standardized factor loadings for each item ranged from 0.690 to 0.845. This suggests that the variable of teacher support demonstrates good reliability and validity.

The parental support dimension was measured by three items (e.g., my parents support my efforts and achievements in studies), measured by agreement level (1 (strongly disagree) to 4 (strongly agree)). A reliability and validity test was performed for these three items, yielding a Cronbach’s alpha coefficient of 0.904. Confirmatory factor analysis results revealed standardized factor loadings ranging from 0.815 to 0.934 (as the confirmatory factor analysis for these three items was based on a saturated model, fit indices are not provided). This indicates good reliability and validity for the variable of parental support.

#### 3.2.2. Mediating Variables

*School belonging.* This variable was measured by six items (e.g., I feel a sense of belonging at school), measured by agreement level (1 (strongly disagree) to 4 (strongly agree)). In assessing the reliability and validity of these six items, the Cronbach’s alpha coefficient was calculated to be 0.809. Confirmatory factor analysis demonstrated a good model fit: X^2^ = 131.477, *df* = 6; CFI = 0.977; TLI = 0.942; RMSEA = 0.084; SRMR = 0.023. Simultaneously, the standardized factor loadings for each item ranged from 0.459 to 0.789. This suggests good reliability and validity for the variable of school belonging.

*Emotional experience.* This study divides emotional experience into two dimensions: positive emotional experience and negative emotional experience. Positive emotional experience was measured by five items (e.g., how often do you feel happy emotions?), measured by frequency (1 (never) to 4 (always)). For the validation of these five items, the Cronbach’s alpha coefficient was found to be 0.824. Confirmatory factor analysis indicated a good model fit: X^2^ = 18.203, *df* = 4; CFI = 0.997; TLI = 0.993; RMSEA = 0.034; SRMR = 0.009. The standardized factor loadings for each item ranged from 0.527 to 0.786, supporting the good reliability and validity of the variable of positive emotional experiences.

Negative emotional experience was measured by four items (e.g., how often do you feel fearful emotions?), also measured by frequency (1 (never) to 4 (always)). A reliability and validity test was conducted for these four items, yielding a Cronbach’s alpha coefficient of 0.768. Confirmatory factor analysis demonstrated a good model fit: X^2^ = 21.078, *df* = 1; CFI = 0.993; TLI = 0.961; RMSEA = 0.082; SRMR = 0.012. Simultaneously, the standardized factor loadings for each item ranged from 0.535 to 0.795, indicating good reliability and validity for the variable of negative emotional experiences.

#### 3.2.3. Dependent Variable

*Resilience.* The resilience variable employed in this study has a unidimensional structure and is measured through five items (e.g., when facing adversity, I can always find a way out) rated on a scale of agreement from 1 (strongly disagree) to 4 (strongly agree). In evaluating the reliability and validity of these five items, the Cronbach’s alpha coefficient was calculated to be 0.782. Confirmatory factor analysis revealed a good model fit: X^2^ = 43.260, *df* = 5; CFI = 0.990; TLI = 0.980; RMSEA = 0.051; SRMR = 0.016. The standardized factor loadings for each item ranged from 0.498 to 0.794, supporting the good reliability and validity of the variable of resilience.

Specifically, due to scientific weighting applied by PISA 2018 to the corresponding items for the study variables, weighted likelihood estimates (WLEs) were ultimately obtained (namely, composite scores). Therefore, this study employed the weighted likelihood estimates corresponding to each variable as the score for that variable in the data analysis. It is important to note that while PISA 2018 did not weigh negative emotions, it did weigh fear of failure. Fear of failure encompasses various negative emotions and represents a negative emotion [70]. Hence, this study employed fear of failure as a proxy variable for negative emotional experience in the data analysis.

### 3.3. Data Analysis

Initially, the data underwent cleansing using SPSS 24, missing data were marked as 555, and descriptive statistics were produced. Subsequently, Mplus 8.3 statistical software was employed to perform the data analysis and construct a mediation model of the variables using path analysis methodology. Finally, the mediation effects were examined using Mplus 8.3.

## 4. Results

### 4.1. Descriptive Statistics and Correlations among the Variables of This Study

The descriptive statistics of the variables are presented in Table 2. From the results, it is evident that disadvantaged students exhibit higher levels of teacher support but lower levels of parental support, school belonging, and resilience indices, while experiencing moderate levels of positive emotional experiences and fear of failure. The correlational analysis indicates a significant negative correlation between fear of failure and other variables, while positive correlations exist significantly among all other variables.

### 4.2. The Effect of Social Support on Resilience: The Chain-Mediated Role of School Belonging and Emotional Experience

Due to the presence of missing data in the dataset used for this study, it was necessary to employ an appropriate approach for handling it. Research suggests that Full Information Maximum Likelihood (FIML) is an effective method for managing missing data [71]. Thus, this study utilized this method for data processing. The standardized path coefficients among the variables are illustrated in Figure 1. From the results, it is observed that teacher support (β = 0.106, *p* < 0.001) and parental support (β = 0.149, *p* < 0.001) significantly positively impact the resilience of disadvantaged students, confirming Hypothesis 1. This suggests that the higher the perceived support from teachers and parents by disadvantaged students, the greater their levels of resilience.

Teacher support (β = 0.163, *p* < 0.001) and parental support (β = 0.255, *p* < 0.001) significantly positively influence the school belonging of disadvantaged students, validating Hypothesis 2. This signifies that increased support from teachers and parents experienced by disadvantaged students corresponds to an enhancement in their sense of school belonging.

School belonging (β = 0.204, *p* < 0.001) significantly predicts the resilience of disadvantaged students, affirming Hypothesis 3. This implies that as the sense of school belonging strengthens among disadvantaged students, their levels of resilience also increase continuously.

Teacher support (β = 0.095, *p* < 0.001) and parental support (β = 0.131, *p* < 0.001) significantly positively impact the positive emotional experiences of disadvantaged students but do not significantly influence fear of failure. Therefore, Hypothesis 5 receives partial confirmation. This indicates that when disadvantaged students receive more support from teachers and parents, their positive emotional experiences strengthen, while the fear of failure remains unchanged.

Positive emotional experiences (β = 0.122, *p* < 0.001) significantly positively influence the resilience of disadvantaged students, while fear of failure (β = −0.129, *p* < 0.001) significantly negatively affects resilience, confirming Hypothesis 6. Therefore, as disadvantaged students experience more positive emotions, their resilience continuously strengthens, whereas experiencing more fear of failure diminishes their resilience levels.

School belonging (β = 0.290, *p* < 0.001) significantly positively affects the positive emotional experiences of disadvantaged students, whereas school belonging (β = −0.238, *p* < 0.001) significantly negatively affects their fear of failure, validating Hypothesis 8. Consequently, when school belonging increases among disadvantaged students, their levels of positive emotional experiences elevate, while their fear of failure decreases.

Based on the aforementioned findings, this study further employed the Bootstrap method to examine the mediating effects, and the standardized results are presented in Table 3. It is observed that school belonging mediates the relationship between teacher support and resilience among disadvantaged students, with an effect size of 0.033 and a 95% confidence interval of [0.0210.046]. Positive emotional experiences also mediate the influence of teacher support on the resilience of disadvantaged students, showing an effect size of 0.012 and a 95% confidence interval of [0.006–0.018]. Furthermore, a chained mediation effect is observed where school belonging and positive emotional experiences mediate the relationship between teacher support and the resilience of disadvantaged students, indicating an effect size of 0.006 and a 95% confidence interval of [0.004–0.008]. Similarly, a chained mediation effect is observed where school belonging and fear of failure mediate the impact of teacher support on the resilience of disadvantaged students, with an effect size of 0.005 and a 95% confidence interval of [0.003–0.007]. The total indirect effect of teacher support on the resilience of disadvantaged students is significant, measuring an effect size of 0.057 with a 95% confidence interval of [0.042–0.073]. As the total effect of teacher support on the resilience of disadvantaged students equals the direct effect plus the indirect effect, the overall effect equals 0.106 + 0.057 = 0.163. The proportion of the indirect effect within the total effect is calculated as 0.057/0.163 = 0.35, signifying that 35% of the effect of teacher support on the resilience of disadvantaged students is mediated by their school belonging, positive emotional experiences, and fear of failure.

Subsequently, school belonging is found to mediate the relationship between parental support and the resilience of disadvantaged students, with an effect size of 0.052 and a 95% confidence interval of [0.035–0.069]. Positive emotional experiences also act as a mediator in the impact of parental support on the resilience of disadvantaged students, with an effect size of 0.016 and a 95% confidence interval of [0.009–0.023]. Additionally, a chained mediation effect is identified where school belonging and positive emotional experiences mediate the relationship between parental support and the resilience of disadvantaged students, indicating an effect size of 0.009 and a 95% confidence interval of [0.006–0.012]. Similarly, a chained mediation effect is observed where school belonging and fear of failure mediate the impact of parental support on the resilience of disadvantaged students, with an effect size of 0.008 and a 95% confidence interval of [0.004–0.011]. The total indirect effect of parental support on the resilience of disadvantaged students is significant, measuring an effect size of 0.087 with a 95% confidence interval of [0.067–0.107]. Since the total effect of parental support on the resilience of disadvantaged students equals the direct effect plus the indirect effect, the overall effect equals 0.149 + 0.087 = 0.236. The proportion of the indirect effect within the total effect is calculated as 0.087/0.236 = 0.37, indicating that 37% of the effect of parental support on the resilience of disadvantaged students is mediated by their school belonging, positive emotional experiences, and fear of failure.

Based on these results, Hypothesis 4 is confirmed, Hypothesis 7 is partially supported, and Hypothesis 9 is validated.

## 5. Discussion

This study employed student data from PISA 2018 across four provinces and municipalities in China to investigate the mechanisms through which social support influences the resilience of disadvantaged students. These research findings provide empirical support for the mediating roles of school belonging, positive emotional experiences, and fear of failure on the relationships between teacher support and parental support and the resilience of disadvantaged students.

### 5.1. The Effect of Social Support on the Resilience of Disadvantaged Students

The results of this study demonstrate that both teacher support and parental support significantly and directly influence the resilience of disadvantaged students. This indicates that when experiencing care and support from teachers and parents, disadvantaged students exhibit enhanced composure and confidence in facing adversity. Continuously breaking through personal barriers, these students gradually elevate their resilience levels throughout this process. This substantiates the protective role of social support in fostering resilience [36,37]. Past research has also indicated that positive family relationships and atmosphere, along with positive feedback from the family, can contribute to the manifestation of resilience behaviors in adolescents [38]. Furthermore, positive teacher–student and peer relationships have been shown to enhance children’s resilience [39]. The process model of resilience posits that individuals, when confronted with challenges, mobilize internal and external protective factors to withstand adversity and stress [72]. Successful adaptation reflects the existing level of an individual’s resilience. Faced with difficulties in life and learning, disadvantaged students leverage external support, such as teacher and parental support, to adeptly manage stress, thereby enhancing their resilience. In other words, for disadvantaged students facing significant changes in their life and learning environments, those with a higher sense of agency internalize external social support or resources as impetus for their development. They effectively harness this support or resources to cope with stress, fostering heightened self-assurance, positive self-appraisal, and an improved capacity for resilience, culminating in favorable psychological adaptation.

### 5.2. The Mediating Role of School Belonging

Based on the aforementioned research outcomes, it is evident that both teacher support and parental support exert positive influences on students’ sense of school belonging. School belonging, in turn, plays a predictive role in fostering resilience among disadvantaged students. It mediates the effects of teacher and parental support on the resilience of disadvantaged students. When disadvantaged students perceive care, encouragement, and support from teachers and parents, their sense of school belonging gradually increases. This leads to a greater acknowledgment and acceptance of teachers, peers, and the school environment, consequently enhancing the resilience levels of disadvantaged students. Previous studies have also indicated that in a supportive and inclusive learning environment, students’ sense of school belonging is satisfied, thereby contributing to the cultivation of positive adaptive skills and resilience in adolescents [51,54]. Drawing from the Triadic Reciprocal Causation theory, human psychology and behavior result from the interaction between external environments and internal factors [73]. School belonging represents a subjective psychological feeling arising from the interaction between individuals and the school environment, fundamentally representing recognition and acceptance within the community [41]. Teachers and parents act as significant figures in students’ socialization processes, serving as bridges between students and the school community. The attitudes displayed by teachers and parents significantly influence students’ subjective perceptions of school. Support from teachers and parents contributes to students feeling accepted and recognized within the school community, thereby fostering the development of a sense of school belonging. Ultimately, school belonging constitutes a positive psychological feeling; students possessing a heightened sense of school belonging experience a sense of value within the collective, resulting in positive emotional experiences of respect and recognition. Positive emotional experiences enhance students’ recognition and evaluation of their own capabilities. These affirmative emotional experiences and successful encounters within the school setting serve as crucial sources for the formation of individual resilience, signifying a critical aspect in the development of resilience.

### 5.3. The Mediating Role of Emotional Experience

From the aforementioned outcomes, it is evident that both teacher and parental support have a positive impact on fostering positive emotional experiences, which, in turn, positively influence the resilience of disadvantaged students. Positive emotional experiences mediate the effects of both teacher and parental support on the resilience of disadvantaged students. The Broaden-and-Build theory of positive emotions posits that positive emotions broaden individuals’ attention and cognitive scope, motivating them to approach and explore unique avenues and establish enduring personal resources to cope with adversity [59]. For disadvantaged students, positive emotions constructively influence their physiological, psychological, and social resources. For example, students with higher levels of positive emotions are more likely to develop resilience [56]. Furthermore, as per the main effects model of social support and the theory of emotional events, understanding social support holds universal benefits: it acts as a critical personal resource, aiding in maintaining an individual’s positive emotional state, thereby reinforcing their confidence in dealing with difficulties, ultimately resulting in more positive emotional experiences [64,74]. Furthermore, research has found that perceived social support by individuals plays a protective role in their psychological adjustment, facilitating an increase in positive emotional experiences [65]. It is evident that positive emotions not only influence the resilience of disadvantaged students, but are also influenced by social support.

However, notably, this study found that social support does not exert a significant impact on the resilience of disadvantaged students through the lens of fear of failure. Instead, it significantly affects resilience through the interaction of school belonging and fear of failure. From this, it is deduced that a lack of social support may not directly cause fear of failure among disadvantaged students; rather, it is the absence of a sense of school belonging that further triggers negative emotional experiences, such as fear of failure. Teachers and parents, as pivotal figures in student socialization, offer students support that fosters feelings of recognition and acceptance, effectively promoting the formation of a sense of belonging [53]. School belonging plays a constructive role in students’ self-concept development, feelings of life significance, and positive evaluations of their abilities, effectively curbing the fear of failure among disadvantaged students [75,76].

### 5.4. The Chain-Mediating Role of School Belonging and Emotional Experience

This study reveals that both teacher and parental support influence the resilience of disadvantaged students through a chained mediating effect of school belonging and positive emotional experiences. Furthermore, teacher and parental support impact the resilience of disadvantaged students through a chained mediating effect of school belonging and fear of failure. On one hand, teacher and parental support effectively foster a sense of school belonging among disadvantaged students. This sense of belonging positively predicts positive emotional experiences, subsequently exerting a positive influence on the resilience of disadvantaged students. On the other hand, teacher and parental support also play a vital role in promoting the sense of school belonging among disadvantaged students. This sense of belonging negatively predicts the feeling of fear of failure, which inversely forecasts the resilience of disadvantaged students. Previous research also suggests that a high level of school belonging enhances positive emotions such as happiness and satisfaction, whereas a low level or absence of belonging is often associated with negative experiences, including depression, anxiety, and loneliness [43,77,78]. Specifically, individuals with a strong sense of school belonging can better handle and confront problems stemming from negative emotions, fostering robust psychological resilience and self-reflection [79,80]. Conversely, if adolescents’ need for school belonging remains unmet, they are more susceptible to negative impacts such as feelings of loneliness and depression [78,81,82]. This underscores the significance of school belonging in aiding disadvantaged students in cultivating positive emotional experiences while inhibiting the adverse effects of negative emotional experiences, ultimately enhancing their resilience. Moreover, favorable teacher and parental support can create a conducive family and school environment for disadvantaged students, enhancing their sense of belonging to the school community. This, in turn, fosters positive emotional experiences, reduces negative emotional experiences, and ultimately facilitates the development of their resilience.

In summary, this study makes a unique contribution by addressing the literature gap on how social support influences resilience among socioeconomically disadvantaged students. While previous research has explored the direct impact of various supportive factors on adolescent resilience, the underlying mediating mechanisms have not been fully revealed. By establishing a sequential mediation model involving school belonging and emotional experiences, this study elucidates how social support influences resilience in disadvantaged students, offering a novel perspective to the research field. It is noteworthy that two crucial factors in the health development of adolescents—school belonging and emotional experiences—have been chosen as mediating variables to investigate the mechanisms influencing resilience in disadvantaged students. This approach provides new insights for subsequent research, as complex relationships in prior studies had not been established.

Furthermore, in terms of the sample population, previous research has predominantly focused on primary and secondary school students, often overlooking the resilience development of students facing socioeconomic and cultural disadvantages. Our study fills this gap by exploring the impact mechanisms of teacher and parental support, two critical external support elements, on the resilience of socioeconomically disadvantaged students. This research presents a new avenue for promoting resilience and healthy development in this particular student demographic.

### 5.5. Limitations and Directions for Future Studies

This study has some limitations that require improvement. Firstly, the data utilized in this study were obtained from secondary school students from four provinces and cities in China via the PISA 2018 survey. It remains unclear whether the analytical outcomes can be generalized to other countries. Therefore, future research could investigate the differences in the impact mechanism of social support on the resilience of disadvantaged students across different countries or cultural backgrounds. Secondly, the study employed cross-sectional data where the establishment of relationships between variables was based on existing theories and prior literature; thus, causality between variables remains uncertain. Subsequent research could employ longitudinal tracking methods to delve deeper into the causal relationships between these variables. Thirdly, the social support considered in this study encompasses only teacher support and parental support, omitting other forms of social support, such as peer support and community support. Consequently, future research should undertake a more comprehensive exploration of diverse aspects of social support and their impact on the resilience of disadvantaged students. Lastly, being constrained by the available data, this study did not delve into additional traits related to resilience, such as social skills and problem-solving abilities, or demographic variables, such as non-alcohol consumption, non-drug use, and participation in extracurricular activities. Therefore, future research studies should collectively include these variables in their considerations to comprehensively investigate the factors influencing the resilience of disadvantaged students.

### 5.6. Implications for Practice

The findings of this study hold significant implications for educational practice. Firstly, it is crucial to prioritize students as the core focus, emphasize the cultivation and development of their resilience, and thereby stimulate disadvantaged student’s initiatives to acquire more resources to cope with adversity. Secondly, teachers and parents should offer positive attention and support to disadvantaged students, establishing an essential social support system that fosters their sense of belonging in school. More importantly, disadvantaged students should be encouraged to perceive and comprehend their relationships with themselves and others. They should have a keen awareness and appreciation for the help they receive from others, transforming this assistance into an intrinsic resource for personal advancement, ultimately fostering the development of their resilience. Thirdly, there is a need to prioritize the psychological well-being of disadvantaged students and maintain and enhance their positive emotional experiences while acquiring positive emotional regulation strategies. Simultaneously, attention should be paid to their negative emotions. In daily life, teachers and parents can provide more encouragement to disadvantaged students, display increased care and communication, understand their learning experiences and difficulties, and offer timely guidance and support. Lastly, reinforcing psychological health education for disadvantaged students is crucial. This education should enhance students’ sense of identification and belonging to the class, guide them to adopt positive and effective emotional regulation strategies to cope with life’s challenges and setbacks, and thereby further enhance their resilience levels.

## 6. Conclusions

Teacher support and parental support serve as external factors facilitating the resilience of disadvantaged students, while school belonging, positive emotional experiences, and fear of failure constitute internal factors influencing resilience. Simultaneously, teacher and parental support indirectly affect the development of resilience in disadvantaged students through their influence on school belonging, positive emotional experiences, and fear of failure. School belonging, positive emotional experiences, and fear of failure function as mediators within this process.

## Figures and Tables

**Figure 1 behavsci-14-00114-f001:**
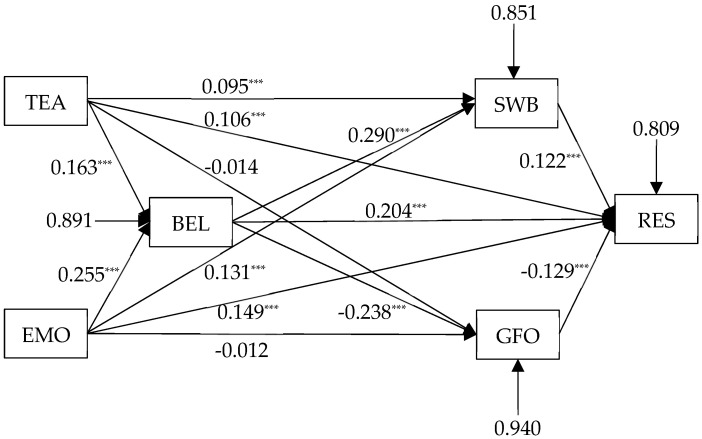
Structural model of relations between social support (teacher support and parental support), school belonging, positive emotional experience, fear of failure and resilience of disadvantaged students. Note: TEA = teacher support; EMO = parental support; BEL = school belonging; SWB = positive emotional experience; GFO = fear of failure; RES = resilience; *** *p* < 0.001.

**Table 1 behavsci-14-00114-t001:** Demographic characteristics of the sample.

Variables	Categories	Frequency (n)	Percentage
Gender	Male	6283	52.1%
	Female	5775	47.9%
School location	A village, hamlet, or rural area	589	4.9%
	A small town	2146	17.8%
	A town	1890	15.7%
	A city	2218	18.4%
	A large city	5215	43.2%
Grade	Grade 7	26	0.2%
	Grade 8	190	1.6%
	Grade 9	4102	34.0%
	Grade 10	7601	63.0%
	Grade 11	132	1.1%
	Grade 12	7	0.1%

**Table 2 behavsci-14-00114-t002:** Means, SDs and correlations for study variables.

Variables	M	SD	Correlations					
			1	2	3	4	5	6
1. TEA	0.31	0.90	—					
2. EMO	−0.21	0.89	0.204 **	—				
3. BEL	−0.31	0.77	0.215 **	0.289 **	—			
4. SWB	0.06	0.88	0.186 **	0.234 **	0.346 **	—		
5. GFO	0.03	0.85	−0.068 **	−0.084 **	−0.245 **	−0.104 **	—	
6. RES	−0.32	0.84	0.211 **	0.268 **	0.346 **	0.257 **	−0.211 **	—

Note: TEA = teacher support; EMO = parental support; BEL = school belonging; SWB = positive emotional experience; GFO = fear of failure; RES = resilience; ** *p* < 0.01.

**Table 3 behavsci-14-00114-t003:** Bootstrap test results of mediation paths.

Path	β	S.E.	95%CI	
			Inf	Sup
TEA→BEL→RES	0.033 ***	0.006	0.021	0.046
TEA→SWB→RES	0.012 ***	0.003	0.006	0.018
TEA→GFO→RES	0.002	0.003	−0.003	0.007
TEA→BEL→SWB→RES	0.006 ***	0.001	0.004	0.008
TEA→BEL→GFO→RES	0.005 ***	0.001	0.003	0.007
TEA→RES (Total indirect effect)	0.057 ***	0.008	0.042	0.073
EMO→BEL→RES	0.052 ***	0.009	0.035	0.069
EMO→SWB→RES	0.016 ***	0.004	0.009	0.023
EMO→GFO→RES	0.002	0.003	−0.004	0.007
EMO→BEL→SWB→RES	0.009 ***	0.002	0.006	0.012
EMO→BEL→GFO→RES	0.008 ***	0.002	0.004	0.011
EMO→RES (Total indirect effect)	0.087 ***	0.010	0.067	0.107

Note: TEA = teacher support; EMO = parental support; BEL = school belonging; SWB = positive emotional experience; GFO = fear of failure; RES = resilience; 95%CI = 95% confidence interval; Inf = Lower limit of 95%CI; Sup = upper limit of 95%CI; *** *p* < 0.001.

## Data Availability

Publicly available datasets were analyzed in this study. These data can be found here: https://www.oecd.org/pisa/data/2018database/ (accessed on 3 December 2023). Our computations of the data are available upon request from the first author.

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
