# Peer review of "How Social Support Affects Resilience in Disadvantaged Students: The Chain-Mediating Roles of School Belonging and Emotional Experience"

_behavsci, 2024, doi:10.3390/bs14020114_

Round 1
Reviewer 1 Report
Comments and Suggestions for Authors
The aim of the present study is to investigate the impact of social support on the resilience of disadvantaged students. The Authors utilize data from the Programme for International Student Assessment 10 (PISA) 2018 conducted in four provinces and cities in China, presenting data from a large sample of students. The topic itself appears interesting and timely, as the mental well-being of the younger members of the population is gaining more and more attention. From a formal perspective, the present article is mostly properly structured. Language quality is good and clear. The Introduction and Discussion sections appear solid and consistent with the purpose of the study, providing the reader with a proper frame to interpret and contextualize the findings. Methods are described rather well, although some minor issues impact on their global presentation. Results are consistent with the intention declared by the Authors and are systematically presented.
While acknowledging the great work done by the Authors and the unquestionable quality of this manuscript, there are some minor observations that they might be willing to address.
1. Lines from 185 o 194, including Table 1, represent a result of the study and should be therefore moved to the proper section accordingly.
2. While recognising the literature as favourable to the hypotheses formulated by the Authors, I suggest citing more recent paper in the literature review. For example, this qualitative paper could be useful in updating the background (DOI: 10.3390/ijerph20054071).
Reviewer 2 Report
Comments and Suggestions for Authors
This manuscript examined the chain-mediated effects
of school belonging and emotional experiences on the relationship between social support and the resilience of disadvantaged students. The authors used data from the Programme for International Student Assessment
(PISA) 2018, conducted in four provinces and cities in China with a sample size of 2997 students. The researchers found that teacher support, parental support, school belonging, and positive emotional
experiences significantly influence the resilience of disadvantaged students, while fear of
failure significantly negatively affects resilience. Additionally, teacher and parental support impacted the resilience of underprivileged students through the chain-mediated effects of school
belonging and positive emotional experiences. Similarly, teacher and parental support also influence
the resilience of disadvantaged students through the chain-mediated effects of school belonging
and fear of failure.
The paper is well-written and organized by the authors. I have a few comments the authors can add to complete the work.
First, in the methods section, the authors have not spelled out the types of variables(e.g., dependent, independent, and mediation). Clarifying the variable types will be of good help to readers.
Second, in the limitation section, the authors should add that social support in the study only focused on parental and teacher support. The authors did not mention other forms of social support, including peers, extended family, and the community.
Also, some demographic variables that are known to influence resilience were not available in the limitations section, including supportive mentors, absence of alcoholism and drug use, close community, and extracurricular activities.
Reviewer 3 Report
Comments and Suggestions for Authors
I have reviewed this article titled “How Social Support Affects Resilience in Disadvantaged 2 Students: The Chain Mediating Role of School Belonging and 3 Emotional Experience”.
1. It is an interesting and very useful study.
2. How the negative effects of parental divorce, parental conflicts, and negativities taking place at school can be avoided?
3. How the authors define the resilience construct?
4. How the disadvantaged students can develop resilience?
5. The study is silent about the age of the students selected as a sample.
6. The study is silent about the type of disadvantage students experienced.
7. The methodology is ok
8. The measurement model is not explained.
9. Other results are fine
10. The discussion can be enhanced by comparing the results of the similar studies.
11. Conclusion is fine
12. Limitations are ok
13. Minor revisions
Reviewer 4 Report
Comments and Suggestions for Authors
There are two main comments I would like the authors to address. First, the article does not clearly state what their contribution is to the current scholarship on the topic. The literature review is done well and captures a thorough understanding of the connection of resilience, support, and belonging. However, the authors essentially suggest their findings are just what other research has stated. There needs to be a paragraph, perhaps in the introduction, that clearly states what is new or how the analysis and findings are pointing at something different or new that needs our attention. Without it, the paper to me reads a bit weak despite its potential.
Second, I wonder if there is a way to be more critical of "resilience" concept. The article suggest that they are looking at the environmental and individual factors that potentially impact resilience development. But it does not critically look at the simplicity, and at times, slippery aspect of resilience as a concept. There's often a sense of deficit-thinking in the employment of "resilience" as a concept as it normalizes individualized "success: despite the odds. However, it is not described in the article how social or structural factors that made students "disadvantaged" in the first place. Understandably, the authors are working within the constraints of the label used in the data-set analyzed. Nonetheless, I think it's a missed opportunity to push against the normalized ways in which research and scholarship reproduces stigmatizing labels imposed on marginalized young people
The article reads well to me, but there are little grammatical mistakes that need to be address through a close copyediting process.
